# Prevention Is Better than Cure—Body Composition and Glycolipid Metabolism after a 24-Week Physical Activity Program without Nutritional Intervention in Healthy Sedentary Women

**DOI:** 10.3390/nu16152536

**Published:** 2024-08-02

**Authors:** Ewa Śliwicka, Natalia Popierz-Rydlewska, Anna Straburzyńska-Lupa, Jivko Nikolov, Łucja Pilaczyńska-Szcześniak, Anna Gogojewicz

**Affiliations:** 1Department of Physiology and Biochemistry, Poznan University of Physical Education, 61-871 Poznań, Poland; 2Department of Food and Nutrition, Poznan University of Physical Education, 61-871 Poznań, Poland; popierz@awf.poznan.pl (N.P.-R.); gogojewicz@awf.poznan.pl (A.G.); 3Department of Physical Therapy and Sports Recovery, Poznan University of Physical Education, 61-871 Poznań, Poland; straburzynskalupa@awf.poznan.pl; 4Department of Geriatrics and Medical Gerontology, Charité Universitätsmedizin Berlin, Hindenburgdamm 30, 12203 Berlin, Germany; jivko.nikolov@charite.de; 5Department of Physiotherapy, Faculty of Medicine and Health Sciences, University of Kalisz, 62-800 Kalisz, Poland; pilaczynskal@gmail.com; 6Department of Radiation Oncology and Radiotherapy, Charité Universitätsmedizin Berlin, Augustenburger Platz 1, 13353 Berlin, Germany

**Keywords:** exercise, nutrition, glucose and lipid metabolism, insulin resistance, adipokines

## Abstract

Women are generally less active than men; therefore, the search for an attractive form of physical activity that benefits women’s health is underway. This study aimed to investigate the influence of a 24-week physical activity program on body composition and indices of carbohydrates and lipid metabolism in sedentary, healthy women. The study comprised 18 female volunteers (mean age 35.0 ± 5.3 years). Dietary intake was assessed using a standardized seven-day food record. Before entering the program and after completing it, each participant’s body composition and indices of glycolipid metabolism were measured. Insulin resistance indexes were calculated based on the obtained data. After the physical activity program, significant decreases in body mass and composition, BMI, waist circumference, percentage of fat content, and fat mass were found. Moreover, there was a significant decrease in glucose, insulin, triglycerides (TG), and resistin concentrations, as well as in the mean values of HOMA-IR and HOMA-AD. A substantial increase in adiponectin levels was also found. To conclude, the combined endurance–resistance physical activity program had a beneficial effect on body mass and composition and improved carbohydrate and lipid metabolism in normal-weight, healthy women. Therefore, we recommend this activity to sedentary young women to prevent obesity and metabolic disorders.

## 1. Introduction

A sedentary lifestyle and poor nutrition are associated with negative health outcomes, such as increased risk of obesity, cardiovascular disease, osteoporosis, premature death, and mental health problems [1,2,3]. Moreover, physical inactivity leads to an altered response to the secretion of myokines and resistance to them, which leads to a pro-inflammatory state that favors sarcopenia and fat accumulation [4].

People are becoming more aware of the benefits of healthful behaviors, such as proper nutrition or regular physical activity, and the enjoyment of practicing physical exercise can positively influence the participants’ well-being [2,3]. The beneficial effects of exercise depend mainly on the type of exercise. Aerobic exercise training instigates changes in anthropometric and cardiometabolic risk factors, including reductions in BMI, body weight, waist circumference, and visceral adipose tissue, and increases in HDL-cholesterol (HDL-Ch) and the maximum rate of oxygen consumption (VO_2_max) [5,6,7]. Meanwhile, resistance training increases lean body mass, reduces exercise-induced oxidative stress in the overweight and those with obesity, and decreases glycated hemoglobin levels in people with abnormal glucose metabolism [8,9]. However, results from network meta-analyses indicate that combined aerobic and resistance exercise is the most effective training for treating and preventing overweight and obesity and type 2 diabetes mellitus (T2DM) [8,9].

Women are generally less physically active than men (34% women versus 29% men) [10], and their bodies tend to have more fat tissue. The content and distribution of fat in women’s bodies change throughout their lives, which can contribute to metabolic risk factors [11]. Therefore, it is crucial to focus on preventing weight gain, particularly visceral fat, to maintain overall health.

Many types of structured and choreographed group physical activities, for example, Zumba, Pilates, Dance Aerobic, Total Body Conditioning, and others, are very popular among women of all ages [12]. Furthermore, a large body of evidence suggests the beneficial effects of aerobics exercise of varying duration and intensity on health indicators in women. However, much of this is limited to overweight and obese individuals and postmenopausal women, as well as exercise combined with nutritional interventions [13,14,15,16,17]. Therefore, the present study aimed to investigate the influence of a 24-week physical activity program in the form of choreographed group fitness training (Dance Aerobic and Total Body Conditioning) on body composition and indices of carbohydrates and lipid metabolism in healthy, sedentary women. Moreover, study participants did not change their diet during the entire physical activity program.

## 2. Materials and Methods

### 2.1. Participants

Initially, the research group consisted of 40 normal-weight women (BMI ≤ 24.9). Due to an attendance rate below 95%, and a lack of 7-day food records and blood samples, the number of participants in the study decreased to 18 volunteers (Figure 1). The women participating in the study were 23–40 years old (mean age 35.0 ± 5.3), declared good health, regularly menstruated, were professionally active, and had not undertaken any physical activity at least 6 months before the research.

For 24 consecutive weeks, the women participated in sixty-minute aerobics classes twice a week, conducted in a fitness club by certified instructors. Before entering the study, all volunteers received detailed information about the research’s purpose and methodology from the research supervisor. Subjects were recommended not to change their lifestyle or nutrition habits during the study period.

The study protocol was reviewed and approved by the Bioethics Committee of the Poznan University of Medical Sciences, reference number 824/10. Participation in the study was voluntary. Participants were informed about the objectives of the study, the details of the procedures, and the possibility of withdrawing at any time. All procedures were carried out in accordance with the ethical standards of the Helsinki Declaration for research involving human subjects, and written informed consent was obtained from all participants before their participation in the study. The characteristics of the participants are presented in Table 1.

### 2.2. Study Design

All procedures were conducted before and after the physical activity program (from October to March), which lasted 24 weeks.

### 2.3. Anthropometric and Body Composition Measurements

All anthropometric measurements were conducted in the morning, in a fasting state, by the same specialist (certified nutritionist). Body mass and height were measured using a certified digital medical-grade scale and a mechanical measuring rod (WPT 60/150.O, Radwag, Radom, Poland), with an accuracy of 0.1 kg for weight and 0.5 cm for height, respectively. Body composition was assessed by the bioimpedance method, using the TANITA BC-420 analyzer (Tokyo, Japan) with GMON Professional software version 3.4.2 (Medizin and Service GmbH, Chemnitz, Germany). Body composition was measured strictly following the recommended measurement conditions [18], as described previously [19,20]. The women were also asked to pay attention to proper hydration because inadequate hydration due to excessive fluid loss or improper fluid intake can result in unreliable results from body composition analysis by bioelectrical impedance analysis (BIA).

### 2.4. Nutritional Assessment

The assessment of dietary intake was conducted using a standardized 7-day food record. The participants were asked to express the number of meals in common measurement units (e.g., glass, cup, bowls, spoons, etc.). The obtained information was adjusted using the album of photographs of food products and dishes elaborated by The National Food and Nutrition Institute in Warsaw (Poland) [21]. Quantitative analysis of the daily food rations’ composition was performed using the Nuvero software package (NUVERO-VOIX, Koszalin, Poland; https://app.nuvero.pl/), which uses a database developed by The National Food and Nutrition Institute in Warsaw [22].

### 2.5. Exercise Energy Expenditure

Average exercise energy expenditure (EEE) was estimated in the first week of the physical activity program, in which women collected their dietary data. EEE was estimated during training sessions, which included Total Body Conditioning (TBC) and Dance Aerobics (DA). During the classes, the Tanita AM-160 Accelerometer (TANITA, Tokyo, Japan) with GMON Professional software version 3.4.2 (Medizin and Service GmbH, Chemnitz, Germany) was used for estimating women’s energy expenditure.

### 2.6. The Physical Activity Program

The physical activity program included strength–endurance classes: Total Body Conditioning (TBC) and Dance Aerobics (DA). A detailed description of the aerobics classes is presented in Figure 1. The classes were held twice a week for 60 min. Each 60-minute class consisted of 10–12 min of warm-up, 40 min of the main activity, and 8–10 min of the final cool-down phase. During TBC sessions, participants targeted the development of all major muscle groups in the body. In the main part of the classes, women performed a series of 8 to 10 repetitions of each exercise. The DA sessions were based on dance movements with constant and moderate intensity. Certified instructors conducted the classes, and the energy expenditure varied based on the intensity of the classes, the musical tempo (measured in beats per minute, BPM), and the inclusion of weight-bearing exercises in TBC classes.

### 2.7. Biochemical Analyses

Blood samples for biochemical analyses were obtained from the antecubital vein in fasting conditions (between 08:00 AM and 10:00 AM). The samples were collected with all safety standards into serum-separating tubes (9 mL, S-Monovette, SARSTEDT, Nümbrecht, Germany) and centrifuged at 2000× *g* for 10 min at 4 °C. The serum was separated from the samples and stored at −70 °C.

In serum, the total cholesterol (cat. No. 7-204), high-density lipoprotein (HDL; cat. No. 7-279), low-density lipoprotein (LDL; cat. No. 7-280), triglycerides (TG, cat. No. 7-253), as well as glucose (cat. No. 7-201) concentrations were analyzed using the Accent 220S automatic biochemical analyzer (Cormay, Łomianki, Poland) and the PZ Cormay S.A. Company reagents (Łomianki, Poland). The serum concentrations of insulin (INS-IRMA, Biosource Europe S.A., Nivelles, Belgium) and adiponectin were measured using immunoradiometric assay kits. Moreover, serum concentrations of hsCRP (DRG International Inc., Springfield Township, NJ, USA), resistin (R&D Systems, Minneapolis, MN, USA), and visfatin (Alpco Diagnostics, Salem, NH, USA) were determined via immuno-enzymatic assay using commercially available kits. Based on the data obtained, the HOMA-IR index was calculated, according to the formula proposed by Matthews [23]:HOMA-IR = fasting insulin (μU/mL) × fasting glucose (mmol/L)/22.5,
and the HOMA-AD index based on the formula of Matsuhisa et al. [24]:HOMA-AD = fasting insulin (mU/L) × fasting glucose (mg/dL)/adiponectin (μg/mL).

The defined cutoff points for different insulin resistance indices used were HOMA-IR ≥ 2.5 [25] and HOMA-AD ≥ 6.26 [26].

### 2.8. Statistical Analyses

All analyses were performed using the Statistica 13.3 software package (TIBCO Software Inc., Palo Alto, CA, USA). Data are presented as means and standard deviations (SD). The Shapiro–Wilk test was used to check the data for normal distribution. In the case of normally distributed variables, the significance of differences between mean values of parameters determined before and after the physical activity program was tested with the Student’s *t*-test. A nonparametric Wilcoxon paired test was used for non-normally distributed variables with inhomogeneous variances. Spearman’s rank analysis was used to calculate correlation coefficients. The level of statistical significance was set at *p* < 0.05.

## 3. Results

### 3.1. Anthropometry and Body Composition

Table 1 presents the results of anthropometric and body composition measurements in women before and after completion of the exercise training. The 24-week exercise training program caused significant decreases in participants’ body mass, BMI, waist circumference, percentage of fat content, and fat mass, and a significant increase in total body water.

### 3.2. Biochemical Indices

As shown in Table 2, the mean values of biochemical indices measured in study participants were within the reference values. After completing the physical activity program, there was a significant decrease in the concentrations of glucose, insulin, TG, and resistin, as well as in the mean values of HOMA-IR and HOMA-AD. A significant increase in adiponectin levels was also found.

### 3.3. Nutritional Evaluation

The average energy values of the diet and selected nutrient intake are presented in Table 3. Obtained nutritional results were compared with recommendations for the Polish population [27]. The diet of the investigated women was assessed by an average energy intake of 1811.8 ± 395.67 kcal·day^−1^ and was consistent with the energy recommendations for women, set at the level of average requirement (EER). The subjects had the proper percentage of macronutrient intake: carbohydrates 45.6 ± 21.07, protein 17.4 ± 4.06, and fat 33.8 ± 9.11. However, a lower than recommended dietary fiber intake was observed.

### 3.4. Correlations

The analysis of correlation was conducted between changes (Δ_1–2_) in variables. Positive associations between waist circumference and insulin (*r* = 0.72; *p* = 0.0007) and resistin (*r* = 0.58; *p* = 0.0125) were found, as well as between HOMA-IR and HOMA-AD (*r* = 0.60; *p* = 0.0085). Visfatin negatively correlated with waist circumference (*r* = −0.47; *p* = 0.0469) and resistin (*r* = −0.65; *p* = 0.0036). A negative association between hsCRP and adiponectin (*r* = −0.56; *p* = 0.0151) was also found.

## 4. Discussion

There is growing evidence that regular physical activity is an important factor that reduces the risk of metabolic diseases [28,29]. In our study, a 24-week endurance–resistance physical activity program, in the form of choreographed group fitness training, contributed to a significant decrease in serum concentrations of hsCRP and resistin, while adiponectin levels increased. These changes were also associated with beneficial changes in body composition and improvements in carbohydrate and lipid metabolism in the women studied.

Adipokines, such as adiponectin, resistin, and visfatin, are produced by adipocytes and other cells of the adipose tissue. They are biologically active autocrine, paracrine, and endocrine substances, which can lead to chronic complications, such as inflammation and insulin resistance [30]. Adiponectin is involved in lipid metabolism, energy regulation, immune response and inflammation, and insulin sensitivity [31]. Several studies in women, mainly overweight/obese subjects, have shown that systematic physical activity modulates adiponectin levels [32,33,34,35]. Our previous study [32] found an increase in adiponectin levels after three-month recreational aqua fitness training in obese women. Racil et al. [33] examined the influence of 12-week interval training of moderate- or high-intensity exercise on adiponectin levels. The authors showed that better modifications in plasma adiponectin levels were achieved through high compared to moderate exercise training intensity in obese young females. In turn, Markofski et al. [35] showed that increased circulating adiponectin levels occurred after 12 weeks of combined aerobic and resistance exercise training without any significant alterations in weight or percentage of body fat content in older, apparently healthy participants. In research conducted by Lakhdar et al. [34], a six-month aerobic exercise alone or combined with diet resulted in a significant increase in circulating adiponectin levels in obese women, independent of changes in body composition.

Resistin has the opposite effect to adiponectin [36]. It promotes inflammation by enhancing the secretion of pro-inflammatory markers, such as tumor necrosis factor-alpha (TNFα) and interleukins [37]. Previous research has indicated that a high resistin level impairs glucose tolerance, increases insulin resistance in the liver, and impairs insulin activity [38]. In this study, we showed decreased resistin levels after an aerobics-based physical activity program in healthy, normal-weight women. There is much research on the impact of exercise on resistin levels [39,40,41,42,43]. Prestes et al. [41] and Botero et al. [43] reported decreased resistin levels in older postmenopausal women after 16 and 12 weeks of resistance training, respectively. Furthermore, Jones et al. [40] demonstrated a significant decrease in serum resistin in obese adolescents after 32 weeks of aerobic exercise. On the contrary, Giannopoulou et al. [39] and Jorge et al. [42] found no significant changes in resistin levels after 12 and 14 weeks of aerobic training in women with T2DM.

Visfatin (nicotinamide phosphoribosyltransferase; NAMPT) exists in two forms: intracellular (iNAMPT) and extracellular (eNAMPT) [44]. Some authors suggested that very high levels of visfatin are associated with increased inflammation, which may contribute to the development of insulin resistance, T2DM, cardiovascular, and renal diseases [45]. We did not observe associations between visfatin levels and insulin resistance in our study for visfatin levels, which remained unchanged. However, its changes (Δ_1–2_) negatively correlated with waist circumference (r = −0.47; *p* = 0.0469). These findings confirmed that circulating visfatin levels are associated with reduced visceral fat in healthy subjects [46]. Moreover, a negative correlation was also found with changes (Δ_1–2_) in resitin levels (*r* = −0.65; *p* = 0.0036).

Changes in adipokine concentrations noted in this study were accompanied by significant changes in body composition, especially in fat mass loss. It should be noted that more favorable changes were observed in women with higher fat stores. Our results are in line with those obtained by other authors [12,47]. Their studies, based on 16-week Zumba Fitness, as well as Zumba Fitness combined with a bodyweight training intervention, demonstrated a significant loss of fat mass and an increase in muscle mass in study participants. In turn, Westerterp et al. [48] indicated that a long-term running training had no effect on body weight in sedentary women and men. The changes in body composition included a loss in fat mass, which was nearly fully compensated by an increase in fat-free mass.

In their systematic review, Yarizadeh et al. [49] indicated that combining both aerobic and resistance training resulted in a lowering of subcutaneous abdominal adipose tissue (SAT), more so than aerobic training or resistance training alone. Although the physical activity program used in our study consisted of aerobic and resistance exercises, we did not observe any changes in FFM. Importantly, women were of normal weight, and according to Westerterp [50], exercise has marginal or no effect on body weight in subjects such as these. We also cannot exclude that the frequency and intensity of the program (twice a week, tempo 110–140 bpm) were insufficient to evoke greater changes in body mass and composition, especially in FFM. Therefore, to achieve more favorable changes in muscle mass, it seems effective to introduce changes in resistance training by increasing intensity [51].

It should be noted that, in addition to exercise, diet also affects weight and body composition. Okura et al. [52] indicated that combined high-intensity dance aerobics and a weight loss diet had an impact on maintaining FFM and reducing the risk of cardiovascular disease. Some authors also suggested that exercise with high protein intake, but without caloric restriction, is effective in achieving a significantly greater lean body mass than exercise alone [53,54]. Study participants consumed appropriate amounts of energy. However, the RDA values for energy intake were established for sedentary women and were sufficient to maintain FFM [54]. Furthermore, protein intake was 1.2 g·kg_bm_^−1^, according to the RDA threshold for the Polish adult population [27], which can prevent muscle mass loss [54]. In normal-weight subjects, exercise had marginal or no effect on body weight. Interestingly, Westerterp et al. [48] suggested that women tend to preserve their energy balance more strongly, compensating for increased energy expenditure with increased energy intake. Therefore, the decrease in body mass and FM in women was significantly lower than that in men.

Regarding our results, we also showed reduced serum insulin and glucose levels and HOMA-IR values with favorable changes in fat mass and fat distribution, which confirmed positive associations between changes (Δ_1–2_) in waist circumference and insulin (r = 0.72; *p* = 0.0007). Similar observations were reported in our previous study [32], in which three-month recreational aqua fitness training caused a significant decrease in serum insulin levels and HOMA-IR values in obese women. Furthermore, Poehlman et al. [55] examined the effects of endurance and resistance training on insulin sensitivity in non-obese young women for six months. The authors showed that both endurance and resistance training improved glucose disposal in young women but through different mechanisms. In addition, Rodziewicz-Flis [56] showed that a 12-week folk dance training performed by older adults significantly improved insulin sensitivity indicators.

In our study, the HOMA-IR and HOMA-AD values decreased significantly (*p* = 0.0000 and *p* = 0.0001, respectively), which indicated an improvement in insulin sensitivity. We also found a positive correlation between these indexes (*r* = 0.60; *p* = 0.0085). These findings confirmed that HOMA-AD is a useful marker to determine insulin resistance [57,58,59].

A large body of evidence exists about the association of physical activity with the blood lipid profile. Systematic exercise leads to increased lipolysis, fatty acid oxidation, and improved mitochondrial function in people with impaired lipid metabolism and healthy people with normal adipose tissue content [60,61]. Most of the results suggest that exercise positively impacts HDL-Ch and TG [62,63,64,65].

The present study showed that a 24-week aerobic–resistance physical activity program decreased TG serum concentrations in healthy women. Our results are in line with those obtained by other authors [66,67,68]. Stasiulis et al. [66] observed significant beneficial changes in blood lipids in young, healthy women undertaking a two-month aerobic cycling training (60 min duration, 3 times a week). On the other hand, Kostrzewa-Nowak et al. [67] reported that a 12-week fitness training program of 2 alternating styles (low and high impact) contributed to a significant decrease in triglycerides and total cholesterol blood concentrations, as well as HDL and LDL cholesterol concentration, in the overweight group. Luo and Zheng [68] examined the influence of a 12-week yoga program combined with aerobic exercise training on morphological and blood lipid indicators in female college students. The authors showed that after completion of the training, there were significant positive changes in HDL-Ch and LDL-Ch in normal-weight, overweight, and obese women, and a significant decrease in TG was observed only in obese participants.

Our study has some potential limitations that should be noted. First, the relatively small sample size. However, this is because the study included only those women whose attendance rate was above 95%, had all anthropometric and biochemical measurements, and completed a seven-day food record. The second is the risk of underestimation and/or inaccurate recording of the products consumed by women [69]. However, the study authors tried to minimize the risk of this fact by educating women about proper diet recording and constant contact with them. Third, we used one of the field methods (BIA) to assess body composition in our study, which is simple, quick, and non-invasive. It should be noted that the obtained results depend on BIA-based predictive equations. Fourth, we did not plan the measurements according to the menstrual cycle of the study participants. This was impossible because all women started the physical activity program on the same day. When using BIA to estimate body composition, it is recommended that women not be tested when they perceive that they retain water during their menstrual cycle. However, the study by Cumberledge et al. [70] indicated that the menstrual cycle phase does not affect the measures of body composition determined by BIA.

## 5. Conclusions

To conclude, a 24-week physical activity program, in the form of choreographed group fitness training, without nutritional intervention, had a beneficial effect on normal-weight, healthy women. It prevented weight gain, especially fat mass, and improved carbohydrate and lipid metabolism. Therefore, we recommend this activity to sedentary young women in prevention of obesity and metabolic disorders.

## Figures and Tables

**Figure 1 nutrients-16-02536-f001:**
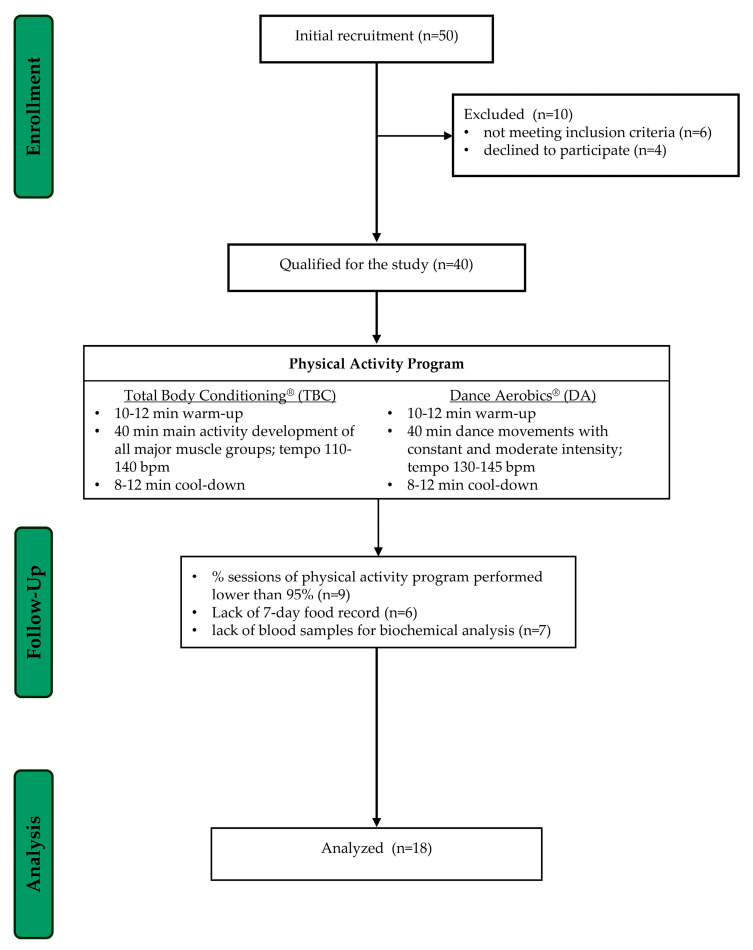
Flowchart of the study selection process.

**Table 1 nutrients-16-02536-t001:** Demographic, anthropometric, and physiological data of the study participants before and after exercise training (n = 18).

Variables	Before	After	*p*-Value
	Mean ± SD	Mean ± SD	
Age (years)	34.7 ± 5.3		
Body height (cm)	163.3 ± 5.5		
Body mass (kg)	61.7 ± 4.5	60.7 ± 4.5	0.0000
BMI	23.2 ± 1.4	22.5 ± 1.6	0.0044
Waist circumference (cm)	83.2 ± 5.7	80.8 ± 5.9	0.0000
Fat (%)	31.5 ± 4.0	30.3 ± 4.4	0.0237
FM (kg)	19.5 ± 3.1	18.5 ± 3.4	0.0047
FFM (kg)	42.3 ± 3.3	42.3 ± 3.0	0.9794
TBW (%)	49.3 ± 3.1	50.1 ± 3.6	0.0200
SBP (mmHg)	117.5 ± 12.6	110.0 ± 10.0	0.0698
DBP (mmHg)	73.8 ± 10.8	71.7 ± 11.9	0.2728

Values are expressed as means ± SD. BMI, body mass index; FM, fat mass; FFM, fat-free body mass; TBW, total body water; SBP, systolic blood pressure; DBP, diastolic blood pressure.

**Table 2 nutrients-16-02536-t002:** Concentrations of selected biochemical indices in the study participants before and after exercise training (n = 18).

Variables	Before	After	*p*-Value
	Mean ± SD	Mean ± SD	
Glucose (mg·dL^−1^)	95.2 ± 8.3	87.9 ± 9.6	0.0006
Insulin (µIU·mL^−1^)	10.4 ± 2.5	8.3 ± 2.4	0.0000
TG (mg·dL^−1^)	98.5 ± 44.5	84.6 ± 28.5	0.0250
Total cholesterol (mg·dL^−1^)	197.0 ± 18.3	193.2 ± 15.5	0.0641
HDL-CH (mg·dL^−1^)	62.0 ± 11.6	62.1 ± 12.8	0.9034
LDL-CH (mg·dL^−1^)	115.4 ± 17.8	114.2 ± 19.4	0.6231
hsCRP (mg·L^−1^)	1.6 ± 2.6	1.1 ± 1.4	0.0494
Resistin (ng·mL^−1^)	5.3 ± 1.8	4.3 ± 1.3	0.0272
Visfatin (ng·mL^−1^)	2.1 ± 1.5	2.1 ± 1.3	0.9467
Adiponectin (µg·mL^−1^)	15.8 ± 2.6	18.2 ± 2.5	0.0000
HOMA-IR	2.43 ± 0.67	1.87 ± 0.69	0.0000
HOMA-AD	3.57 ± 1.32	2.26 ± 0.90	0.0001

Values are expressed as means ± SD. TG, triglycerides; HDL-CH, high-density lipoprotein cholesterol; LDL-CH, low-density lipoprotein cholesterol.

**Table 3 nutrients-16-02536-t003:** Energy intake and selected nutrients’ intake in the study group.

Nutrient	Mean ± SD	RDA
Energy intake (kcal)	1811.8 ± 395.7	1800–1900
EEE during DA (kcal)	331.1 ± 122.6	
EEE during TBC (kcal)	353.6 ± 61.0	
CHO		
(g)	230.6 ± 75.8	
(% energy)	45.6 ± 21.1	45–65
(g·kg_bm_^−1^)	3.7 ± 1.1	
Saccharose (g)	42.8 ± 19.4	
Saccharose (% energy)	9.2 ± 3.3	
Fiber (g)	18.3 ± 6.6	25 g
PRO		
(g)	76.9 ± 16.2	
(% energy)	17.4 ± 4.1	10–20
(g·kg_bm_^−1^)	1.2 ± 0.2	0.9
FAT		
(g)	66.8 ± 20.5	
(% energy)	33.8 ± 9.1	20–35
(g·kg_bm_^−1^)	1.3 ± 0.3	0.5–1.0
Cholesterol (mg)	252.4 ± 73.2	<300
SFA (g) SFA (%)	25.4 ± 6.4 13.0 ± 3.3	
PUFA (g) PUFA (%)	8.2 ± 2.6 4.1 ± 1.11	0.5–1.0 ^§^ 6–10
MUFA (g) MUFA (%)	24.3 ± 6.83 12.3 ± 3.19	

Values are expressed as means ± SD. EEE, exercise energy expenditure; DA, Dance Aerobics; TBC, Total Body Conditioning; CHO, carbohydrates; PRO, proteins; FAT, lipids; SFA, saturated fatty acids; PUFA, polyunsaturated fatty acids; MUFA, monounsaturated fatty acids. ^§^ American Heart Association, American Dietetic of Canada.

## Data Availability

The data supporting the reported results are available upon request from the corresponding author (E.Ś.) due to privacy.

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
