# Peer review of "Prevention Is Better than Cure—Body Composition and Glycolipid Metabolism after a 24-Week Physical Activity Program without Nutritional Intervention in Healthy Sedentary Women"

_nutrients, 2024, doi:10.3390/nu16152536_

Round 1
Reviewer 1 Report
Comments and Suggestions for Authors
Nice study, no rocket science. To many drop-outs: 95% to stringent? No control group. No info on behaviour during the study (increase in activity and nutritional intake).
Some additional remarks to improve the paper.
Abstract
-Please make clear why the study is important.
-The phrasing "concentrations of biochemical indices" is a little bit odd/ PLease change
Introduction
Please make clearer: in the last lines of the intro the aim(s) of the study becomes clear (choreographed group fitness training) and apparently in healthy women?
Materials and methods
-Please include a chart that details how 22 women did not reach the final analyses.
-Were there postmenopausal women in the study? Were measurements planned according to the menstrual cycle (mid follicular phase?!)?
-Why was there no control group?
-I would rather mention that EEE was estimated and not assessed. Or provide reliability/validation for the accelerometers.
Results
-Personally, I do not think that p values must be reported in text and tables (e.g. glucoe, insulin etc).
-Was nutritional intake not collected before and after?
-I am not sure what the correlations add to the paper. There were no corrections for multiple testing.
Discussion
Apparently (line 340-341) women were nutritionally supported throughout the study: thgis should be in the methods section.
Author Response
25.07.2024
We present our responses to the Reviewer's remarks below.All suggestions were very helpful for us and have been now incorporated in the revised manuscript. On behalf of all co-authors I would like to clarify the points raised by the Reviewer. I hope that the Reviewer and Editors will be satisfied with the responses to their comments and will recognize the re-edited manuscript as acceptable for publication in Nutrients.
All instructions are taken into account and corrected in the text. The answers are given in the appropriate sections below.
Abstract
Please make clear why the study is important.
The phrasing "concentrations of biochemical indices" is a little bit odd/ Please change
Thank you for these suggestions.
We changed the title:
„Prevention is better than cure - body composition and glycolipid metabolism after a 24-week physical activity program without nutritional intervention in healthy sedentary women.”
We made some corrections in the abstract.
„Women are generally less active than men, therefore, the search for an attractive form of physi-cal activity that benefits women's health is underway. The study aimed to investigate the influence of a 24-week physical activity program on body composition and indices of carbohydrates and lipid metabolism in sedentary, healthy women.”
Introduction
Please make clearer: in the last lines of the intro the aim(s) of the study becomes clear (choreographed group fitness training) and apparently in healthy women?
Thank you very much for this comment. This section has been rewritten:
„Women are generally less physically active than men (34% women versus 29% men) [10], and their bodies tend to have more fat tissue. The content and distribution of fat in women’s bodies change throughout their lives, which can contribute to metabolic risk fac-tors [11]. Therefore, it is crucial to focus on preventing weight gain, particularly visceral fat, to maintain overall health.
Many types of structured and choreographed group physical activities, for example, Zumba®, Pilates®, Dance Aerobic®, Total Body Conditioning®, and others, are very popular among women of all ages [12]. Furthermore, a large body of evidence suggests the beneficial effects of aerobics exercise of varying duration and intensity on health indicators in women. However, much of this is limited to overweight and obese individuals and post-menopausal women, as well as exercise combined with nutritional interventions [13–17]. Therefore, the present study aimed to investigate the influence of a 24-week physical activity program in the form of choreographed group fitness training (Dance Aerobic®, Total Body Conditioning®) on body composition and indices of carbohydrates and lipid metab-olism in healthy, sedentary women. Moreover, study participants did not change their diet during the entire physical activity program.”
Materials and methods
Please include a chart that details how 22 women did not reach the final analyses.
Thank you very much for this suggestion. The flow chart and a detailed description of study participants have been added.
„Initially, the research group consisted of 40 women. Due to an attendance rate below 95%, and a lack of 7-day food records and blood samples, the number of participants in the study decreased to 18 volunteers (Figure 1).”
Were there postmenopausal women in the study?
There were no postmenopausal women in the study. We added this information in the text.
„The women participating in the study were 23-40 years old (mean age 35.0 ± 5.3), declared good health, regularly menstruated, were professionally active, and had not undertaken any physical activity at least 6 months before the research.”
Were measurements planned according to the menstrual cycle (mid follicular phase?!)?
All women started the physical activity program on the same day, so we did not plan the measurements according to the menstrual cycle. However, in the limitations section we added information about the potential influence of the menstrual cycle on body composition in women:
„Fourth, we did not plan the measurements according to the menstrual cycle of the study participants. It was impossible because all women started the physical activity program on the same day. Using BIA to estimate body composition, it is recommended that women not be tested when they perceive that they retain water during their menstrual cycle. However, the study by Cumberledge et al. [70] indicates that the menstrual cycle phase does not affect the measures of body composition determined by BIA.”
Why was there no control group?
There was a control group, but only in the first term of the study. Therefore, we decided not to include it in this study.
I would rather mention that EEE was estimated and not assessed. Or provide reliability/validation for the accelerometers.
Thank you very much for this comment. We provided changes as suggested by the Reviewer.
„Average exercise energy expenditure (EEE) was estimated in the first week of physical activity program, in which women collected their dietary data. EEE was estimated during training sessions, which included Total Body Conditioning® (TBC) and Dance Aerobics® (DA).”
Results
Personally, I do not think that p values must be reported in text and tables (e.g. glucoe, insulin etc).
We removed p-values from the text as suggested by the Reviewer.
Was nutritional intake not collected before and after?
We collected nutritional intake only at the beginning of the study.
I am not sure what the correlations add to the paper.
We decided not to remove them from the paper.
There were no corrections for multiple testing.
In our opinion, the study is restricted to a rather small number of comparisons. Furthermore, if multiple usages of a simple test such as 't', and if the results of the individual tests are important, the exact p-values for each test should be quoted and discussed appropriately.
Discussion
Apparently (line 340-341) women were nutritionally supported throughout the study: thgis should be in the methods section.
This information is included in the Materials and methods section:
Lines 84-85: „Subjects were recommended not to change their lifestyle or nutrition habits during the study period.”
Reviewer 2 Report
Comments and Suggestions for Authors
Thank you for submitting the manuscript "The effect of combined endurance-resistance physical activity program without nutritional intervention on body composition and metabolic indices in healthy women" to Nutrients. The manuscript is very well written and the researchers conducted standard physical training with women and subjected these individuals to physical, biochemical and anthropometric assessments. I have some suggestions to improve the quality of the manuscript:
- authors need to check the similarity report as the percentage is high for a research article.
- Line#27: body mass and BMI seem to talk about the same thing.
- My suggestion is to include numerical values followed by probability values in the abstract.
- Line#70: prior to how long?
- It seems to me that table 1 shows that there was no gain in lean mass, only a decrease in fat mass, which resulted in a difference in body mass between the beginning and end of the study. As the group was already classified in the BMI of eutrophy, the authors can better discuss why the evaluated group did not increase lean mass (expected).
Author Response
25.07.2024
We present our responses to the Reviewer's remarks below. All suggestions were very helpful for us and have been now incorporated in the revised manuscript. On behalf of all co-authors I would like to clarify the points raised by the Reviewer. I hope that the Reviewer and Editors will be satisfied with the responses to their comments and will recognize the re-edited manuscript as acceptable for publication in Nutrients.
All instructions are taken into account and corrected in the text. The answers are given in the appropriate sections below.
Authors need to check the similarity report as the percentage is high for a research article.
Thank you for this comment. We rephrased or changed the places highlighted in the report.
Line#27: body mass and BMI seem to talk about the same thing.
We understand your comment; however, we disagree with this statement. Body mass index (BMI) is the method of using an adult's height and weight to broadly place them into underweight, normal weight, overweight, and obese categories. we used it to qualify women for the study. We added this information in the Materials and methods section.
„Initially, the research group consisted of 40 normal weight women (BMI ≤24.9).”
My suggestion is to include numerical values followed by probability values in the abstract.
Although we agree with your suggestion, due to the limited words' number in the abstract (200) it is impossible to show all numerical values followed by probability values.
Line#70: prior to how long?
We specified the minimum period before the study during which women did not engage in any physical activity. We added this information in the Materials and methods section.
„The women participating in the study were 23-40 years old (mean age 35.0 ± 5.3), declared good health, regularly menstruated, were professionally active, and had not undertaken any physical activity at least 6 months before the research.”
It seems to me that table 1 shows that there was no gain in lean mass, only a decrease in fat mass, which resulted in a difference in body mass between the beginning and end of the study. As the group was already classified in the BMI of eutrophy, the authors can better discuss why the evaluated group did not increase lean mass (expected).
Thank you for this comment. We tried to justify our results in the text. We discuss not only the influence of exercise on body mass and composition but also diet.
“Changes in adipokine concentrations noted in this study were accompanied by significant changes in body composition, especially in fat mass loss. It should be noted that more favorable changes were observed in women with higher fat stores. Our results are in line with those obtained other authors [12,47]. Their studies based on 16-week Zumba Fitness®, as well as Zumba Fitness® combined with bodyweight training intervention, have demonstrated a significant loss of fat mass and increase in muscle mass in study participants. In turn, Westerterp et al. [48] indicated that a long-term running training, had no effect on body weight in sedentary women and men. The changes in body composition included a loss in fat mass, which was nearly fully compensated by an increase in fat-free mass.
In their systematic review, Yarizadeh et al. [49] indicated that combining both aerobic and resistance training resulted in a lowering of subcutaneous abdominal adipose tissue (SAT) more than aerobic training or resistance training alone. Although the physical activity program used in our study consisted of aerobic and resistance exercises, we did not observe any changes in FFM. Importantly, women were normal-weight, and according to Westerterp [50], exercise has marginal or no effect on body weight in subjects like these. We also cannot exclude that the frequency and intensity of the program (twice a week, tempo 110-140 bpm) were insufficient to evoke greater changes in body mass and composition, especially in FFM. Therefore, to achieve more favorable changes in muscle mass, it seems effective to introduce changes in resistance training by increasing intensity [51].
It should be noted that, in addition to exercise, diet also affects weight and body com-position. Okura et al. [52] indicated that combined high-intensity dance aerobics and a weight loss diet had an impact on maintaining FFM and reducing the risk of cardiovascular disease. Some authors also suggested that exercise with high protein intake, but without caloric restriction, is effective in achieving significantly greater lean body mass than exercise alone [53,54]. Study participants consumed appropriate amounts of energy. However, the RDA values for energy intake are established for sedentary women and were sufficient to maintain FFM [54]. Furthermore, protein intake was 1.2 g · kgbm-1, according to the RDA threshold for the Polish adult population [27], which can prevent muscle mass loss [54]. In normal-weight subjects, exercise had marginal or no effect on body weight . Interestingly, Westerterp et al. [48] suggested that women tend to preserve their energy balance more strongly, compensating for increased energy expenditure with increased energy intake. Therefore, the decrease in body mass and FM in women is significantly lower than that of men.”
Reviewer 3 Report
Comments and Suggestions for Authors
Thank you for giving me the opportunity to review this study. I have found it very interesting, especially the length of the intervention, which renders this study important. Still, the manuscript can be improved. I have a few suggestions, as outline below:
Abstract
L23 add “y” for years in “35 ± 5.3 y”. Also the number of decimal points should be the same. Age could have zero or one decimal point.
Maybe the authors could have less information on insulin resistance measurements, and add some information on the modality of the exercise program.
Some numerical values of the results are needed in the abstract.
Introduction
L47-49 “Regarding….Mellitus”, please check syntax
L60-61 “in who physical inactivity is more prevalent than men (34% versus 60 29 %)”, please check syntax
The positives of independent endurance or resistance exercise are mentioned. However there is nott enough information on the outcomes (benefits or lack of) of concurrent aerobic+resistance training. Some more information on concurrent training would improve the introduction.
Methods
L69 add years and check decimal points
L128 the word “measured” is wrong, as EEE was estimated. Please rephrase, or connect with the following sentence L132-133
Table 1 and 2 and 3. Decimal points for values should be in accordance to the sensitivity of the various methods used. Mean and SD should have the same decimal points. Based on the methodology used, I believe having one decimal point. I.e. Body mass 61.7 ± 4.6, waist circumference 83.2 ± 5.7
Discussion
L238-239 The changes on body composition were statistically significant, but they were also marginal. What is the clinical significance of such small changes? Please discuss the clinical significance of the findings as numerical changes are very small.
L295 – 301 Although the information might be correct, this paragraph is not associated with anything measured in the present study. Maybe delete it or connect use the space to write more about your own measurements and findings.
The conclusion is too strong. Although correct based on the numerical results, I believe that in essence the exercise program used did not do much for overall fitness. The participants lost 1 kg over 24 weeks, which apparently was fat, since FFM did not change over time. So the resistance training used was not enough to elicit any FFF increases.
Overall, the fact that this study lasted 24 weeks is very important. The tile overemphasizes the combination of endurance and resistance, something that was not clearly displayed in the manuscript, since there is no detailed description of the exercise modality, and especially intensity. The biochemical findings are important. Still, frequency of exercise was 2 times per week, there was no control group, the exercise protocol is not described in detail. I believe this study is important, because it shows that exercising two times per week consistently can offer some benefits, but on the other hand the clinical improvements are overestimated in the present study. Make some changes in the discussion to show tha importance of exercise, maybe mention tha exercising two-times per week without controlling diet could offer some benefit, but obviously more (either frequency or intensity or workload) is required for better results.
Comments on the Quality of English LanguageGood quality, I have only found a few syntax errors.
Author Response
25.07.2024
We present our responses to the Reviewer's remarks below. All suggestions were very helpful for us and have been now incorporated in the revised manuscript. On behalf of all co-authors I would like to clarify the points raised by the Reviewer. I hope that the Reviewer and Editors will be satisfied with the responses to their comments and will recognize the re-edited manuscript as acceptable for publication in Nutrients.
All instructions are taken into account and corrected in the text. The answers are given in the appropriate sections below.
Abstract
L23 add “y” for years in “35 ± 5.3 y”. Also the number of decimal points should be the same. Age could have zero or one decimal point.
Thank you for this comment. We made changes in the text:
„The study comprised 18 female volunteers (mean age 35.0 ± 5.3 y).”
Maybe the authors could have less information on insulin resistance measurements, and add some information on the modality of the exercise program.
Some numerical values of the results are needed in the abstract.
Although we agree with your suggestion, due to the limited words' number in the abstract (200) it is impossible to show numerical values followed by probability values.
Introduction
L47-49 “Regarding….Mellitus”, please check syntax
L60-61 “in who physical inactivity is more prevalent than men (34% versus 60 29 %)”, please check syntax
The positives of independent endurance or resistance exercise are mentioned. However there is nott enough information on the outcomes (benefits or lack of) of concurrent aerobic+resistance training. Some more information on concurrent training would improve the introduction.
Methods
L69 add years and check decimal points
We made changes in the text as suggested by the Reviewer.
„The women participating in the study were 23-40 years old (mean age 35.0 ± 5.3), declared good health, regularly menstruated, were professionally active, and had not undertaken any physical activity at least 6 months before the research.”
L128 the word “measured” is wrong, as EEE was estimated. Please rephrase, or connect with the following sentence L132-133
Thank you very much for this comment. We provided changes as suggested by the Reviewer.
„Average exercise energy expenditure (EEE) was estimated in the first week of physical activity program, in which women collected their dietary data. EEE was estimated during training sessions, which included Total Body Conditioning® (TBC) and Dance Aerobics® (DA).”
Table 1 and 2 and 3. Decimal points for values should be in accordance to the sensitivity of the various methods used. Mean and SD should have the same decimal points. Based on the methodology used, I believe having one decimal point. I.e. Body mass 61.7 ± 4.6, waist circumference 83.2 ± 5.7
We corrected decimal points in Table 1, 2 and 3 as suggested by the Reviewer.
Discussion
L238-239 The changes on body composition were statistically significant, but they were also marginal. What is the clinical significance of such small changes? Please discuss the clinical significance of the findings as numerical changes are very small.
Some sections in the discussion have been rewritten:
“Changes in adipokine concentrations noted in this study were accompanied by significant changes in body composition, especially in fat mass loss. It should be noted that more favorable changes were observed in women with higher fat stores. Our results are in line with those obtained other authors [12,47]. Their studies based on 16-week Zumba Fitness®, as well as Zumba Fitness® combined with bodyweight training intervention, have demonstrated a significant loss of fat mass and increase in muscle mass in study participants. In turn, Westerterp et al. [48] indicated that a long-term running training, had no effect on body weight in sedentary women and men. The changes in body composition included a loss in fat mass, which was nearly fully compensated by an increase in fat-free mass.
In their systematic review, Yarizadeh et al. [49] indicated that combining both aerobic and resistance training resulted in a lowering of subcutaneous abdominal adipose tissue (SAT) more than aerobic training or resistance training alone. Although the physical activity program used in our study consisted of aerobic and resistance exercises, we did not observe any changes in FFM. Importantly, women were normal-weight, and according to Westerterp [50], exercise has marginal or no effect on body weight in subjects like these. We also cannot exclude that the frequency and intensity of the program (twice a week, tempo 110-140 bpm) were insufficient to evoke greater changes in body mass and composition, especially in FFM. Therefore, to achieve more favorable changes in muscle mass, it seems effective to introduce changes in resistance training by increasing intensity [51].
It should be noted that, in addition to exercise, diet also affects weight and body com-position. Okura et al. [52] indicated that combined high-intensity dance aerobics and a weight loss diet had an impact on maintaining FFM and reducing the risk of cardiovascular disease. Some authors also suggested that exercise with high protein intake, but without caloric restriction, is effective in achieving significantly greater lean body mass than exercise alone [53,54]. Study participants consumed appropriate amounts of energy. However, the RDA values for energy intake are established for sedentary women and were sufficient to maintain FFM [54]. Furthermore, protein intake was 1.2 g · kgbm-1, according to the RDA threshold for the Polish adult population [27], which can prevent muscle mass loss [54]. In normal-weight subjects, exercise had marginal or no effect on body weight . Interestingly, Westerterp et al. [48] suggested that women tend to preserve their energy balance more strongly, compensating for increased energy expenditure with increased energy intake. Therefore, the decrease in body mass and FM in women is significantly lower than that of men.”
L295 – 301 Although the information might be correct, this paragraph is not associated with anything measured in the present study. Maybe delete it or connect use the space to write more about your own measurements and findings.
Thank you for this comment. We removed this paragraph from the paper.
The conclusion is too strong. Although correct based on the numerical results, I believe that in essence the exercise program used did not do much for overall fitness. The participants lost 1 kg over 24 weeks, which apparently was fat, since FFM did not change over time. So the resistance training used was not enough to elicit any FFF increases.
Overall, the fact that this study lasted 24 weeks is very important. The tile overemphasizes the combination of endurance and resistance, something that was not clearly displayed in the manuscript, since there is no detailed description of the exercise modality, and especially intensity. The biochemical findings are important. Still, frequency of exercise was 2 times per week, there was no control group, the exercise protocol is not described in detail. I believe this study is important, because it shows that exercising two times per week consistently can offer some benefits, but on the other hand the clinical improvements are overestimated in the present study. Make some changes in the discussion to show tha importance of exercise, maybe mention tha exercising two-times per week without controlling diet could offer some benefit, but obviously more (either frequency or intensity or workload) is required for better results.
Thank you very much for these comments.
We changed the title:
„Prevention is better than cure - body composition and glycolipid metabolism after a 24-week physical activity program without nutritional intervention in healthy sedentary women.”
We added Figure 1. with flowchart of the study selection proces and detailed description of physical activity program
We made some improvements in the conclusions:
„To conclude, a 24-week physical activity program, in the form of choreographed group fitness training, without nutritional intervention, has a beneficial effect on normal-weight, healthy women. It prevents weight gain, especially fat mass and improves carbohydrate and lipid metabolism. Therefore, we recommend this activity to sedentary young women in prevention of obesity and metabolic disorders.”
Round 2
Reviewer 3 Report
Comments and Suggestions for Authors
Very good work
The manuscript is better to read now, my compliments to the authors